# Trends in Net Survival from Vulvar Squamous Cell Carcinoma in Italy (1990–2015)

**DOI:** 10.3390/jcm12062172

**Published:** 2023-03-10

**Authors:** Silvia Mancini, Lauro Bucchi, Federica Zamagni, Flavia Baldacchini, Emanuele Crocetti, Orietta Giuliani, Alessandra Ravaioli, Rosa Vattiato, Mario Preti, Rosario Tumino, Stefano Ferretti, Annibale Biggeri, Paola Ballotari, Lorenza Boschetti, Angelita Brustolin, Adele Caldarella, Rossella Cavallo, Claudia Cirilli, Annarita Citarella, Maria L. Contrino, Luigino Dal Maso, Rosa A. Filiberti, Mario Fusco, Rocco Galasso, Fernanda L. Lotti, Michele Magoni, Lucia Mangone, Giuseppe Masanotti, Guido Mazzoleni, Walter Mazzucco, Anna Melcarne, Maria Michiara, Paola Pesce, Angela Pinto, Daniela Piras, Roberto V. Rizzello, Magda Rognoni, Stefano Rosso, Massimo Rugge, Giuseppe Sampietro, Santo Scalzi, Tiziana Scuderi, Giovanna Tagliabue, Federica Toffolutti, Susanna Vitarelli, Fabio Falcini

**Affiliations:** 1Romagna Cancer Registry, Romagna Cancer Institute (IRCCS Istituto Romagnolo per lo Studio dei Tumori (IRST) Dino Amadori), 47014 Meldola, Italy; silvia.mancini@irst.emr.it (S.M.); lauro.bucchi@irst.emr.it (L.B.); flavia.baldacchini@irst.emr.it (F.B.); emanuelecrocetti@yahoo.com (E.C.); orietta.giuliani@irst.emr.it (O.G.); alessandra.ravaioli@irst.emr.it (A.R.); rosa.vattiato@irst.emr.it (R.V.); fabio.falcini@irst.emr.it (F.F.); 2Department of Obstetrics and Gynaecology, University of Torino, 10124 Torino, Italy; mario.preti@unito.it; 3Cancer Registry and Histopathology Department, Provincial Health Authority (ASP), 97100 Ragusa, Italy; rosario.tumino@asp.rg.it; 4Department of Morphology, Surgery and Experimental Medicine, University of Ferrara, Local Health Authority, 44121 Ferrara, Italy; stefano.ferretti@unife.it; 5Unit of Biostatistics, Epidemiology and Public Health, Department of Cardiac, Thoracic, Vascular Sciences and Public Health, University of Padua, 35128 Padua, Italy; annibale.biggeri@ubep.unipd.it; 6Mantova & Cremona Cancer Registry, Epidemiology Unit, Val Padana Health Protection Agency, 46100 Mantova, Italy; paola.ballotari@ats-valpadana.it; 7Pavia Cancer Registry, Public Health Agency of Pavia, 27100 Pavia, Italy; lorenza_boschetti@ats-pavia.it; 8Unit of Epidemiology and Cancer Registry, Local Health Authority, 01100 Viterbo, Italy; 9Tuscany Cancer Registry, Clinical and Descriptive Epidemiology Unit, Institute for Cancer Research, Prevention and Clinical Network (ISPRO), 50139 Florence, Italy; a.caldarella@ispro.toscana.it; 10Cancer Registry of Local Health Authority Salerno, 84124 Salerno, Italy; ro.cavallo@aslsalerno.it; 11Modena Cancer Registry, Public Health Department, Local Health Authority, 41126 Modena, Italy; c.cirilli@ausl.mo.it; 12Cancer Registry, Department of Prevention, Unit of Epidemiology, Local Health Authority, 82100 Benevento, Italy; dp.sep@aslbenevento1.it; 13Cancer Registry of the Province of Siracusa, 96100 Siracusa, Italy; rtp@asp.sr.it; 14Cancer Epidemiology Unit, Centro di Riferimento Oncologico di Aviano (CRO) IRCCS, 33081 Aviano, Italy; dalmaso@cro.it (L.D.M.); federica.toffolutti@cro.it (F.T.); 15Liguria Cancer Registry, IRCCS Ospedale Policlinico San Martino, 16132 Genova, Italy; rosa.filiberti@hsanmartino.it; 16Napoli 3 Sud Cancer Registry, 80031 Napoli, Italy; mariofusco2@virgilio.it; 17Unit of Regional Cancer Registry, Clinical Epidemiology and Biostatistics, IRCCS-CROB, Basilicata, 85028 Rionero in Vulture, Italy; rocco.galasso@crob.it; 18Brindisi Cancer Registry, Local Health Authority, 72100 Brindisi, Italy; fernandalotti@libero.it; 19Cancer Registry of Brescia Province, Epidemiology Unit, Brescia Health Protection Agency, 25124 Brescia, Italy; michele.magoni@ats-brescia.it; 20Epidemiology Unit, Azienda Unità Sanitaria Locale–IRCCS di Reggio Emilia, Via Amendola 2, 42122 Reggio Emilia, Italy; mangone.lucia@ausl.re.it; 21Section of Public Health and RTUP Register, Department of Experimental Medicine, University of Perugia, 06123 Perugia, Italy; giuseppe.masanotti@unipg.it; 22South Tyrol Tumor Registry, 39100 Bolzano, Italy; guido.mazzoleni@asbz.it; 23Department of Health Promotion, Maternal and Infant Care, Internal Medicine and Medical Specialties (PROMISE), University of Palermo, 90131 Palermo, Italy; walter.mazzucco@unipa.it; 24Lecce Province Cancer Registry, 73100 Lecce, Italy; melcarneanna@gmail.com; 25Parma Cancer Registry, Medical Oncology Unit, University Hospital of Parma, 43126 Parma, Italy; michiara@ao.pr.it; 26Catania, Messina and Enna Cancer Registry, 95123 Catania, Italy; paolapesce@tiscali.it; 27Barletta, Andria, Trani Cancer Registry, BAT Province, 76121 Barletta, Italy; pintoangela74@gmail.com; 28Sassari Cancer Registry, Azienda Regionale per la Tutela della Salute-ATS, 7100 Sassari, Italy; dani.piras@atssardegna.it; 29Trento Province Cancer Registry, Unit of Clinical Epidemiology, Azienda Provinciale per i Servizi Sanitari (APSS), 38123 Trento, Italy; roberto.rizzello@apss.tn.it; 30Epidemiology Unit, Cancer Registry of ATS Brianza, Health Protection Agency, 20900 Monza, Italy; magda.rognoni@ats-brianza.it; 31Piedmont Cancer Registry, A.O.U. Città della Salute e della Scienza di Torino, 10123 Turin, Italy; stefano.rosso@cpo.it; 32Veneto Tumour Registry, Azienda Zero, University of Padova-DIMED, 35132 Padova, Italy; massimo.rugge@unipd.it; 33Bergamo Cancer Registry, ATS Bergamo, 24121 Bergamo, Italy; giuseppe.sampietro@ats-bg.it; 34Catanzaro ASP Cancer Registry, 88100 Catanzaro, Italy; scalzi57@gmail.com; 35Trapani and Agrigento Cancer Registry, 91100 Trapani, Italy; trusyit@yahoo.it; 36Lombardy Cancer Registry-Varese Province, Cancer Registry Unit, Department of Research, Fondazione IRCCS Istituto Nazionale Tumori, 20133 Milan, Italy; giovanna.tagliabue@istitutotumori.mi.it; 37Macerata Province Cancer Registry, University of Camerino, 62032 Camerino, Italy; susanna.vitarelli@unicam.it; 38Cancer Prevention Unit, Local Health Authority, 47121 Forlì, Italy

**Keywords:** vulvar neoplasms, survival, trend

## Abstract

(1) Objective: In many Western countries, survival from vulvar squamous cell carcinoma (VSCC) has been stagnating for decades or has increased insufficiently from a clinical perspective. In Italy, previous studies on cancer survival have not taken vulvar cancer into consideration or have pooled patients with vulvar and vaginal cancer. To bridge this knowledge gap, we report the trend in survival from vulvar cancer between 1990 and 2015. (2) Methods: Thirty-eight local cancer registries covering 49% of the national female population contributed the records of 6274 patients. Study endpoints included 1- and 2-year net survival (NS) calculated using the Pohar-Perme estimator and 5-year NS conditional on having survived two years (5|2-year CNS). The significance of survival trends was assessed with the Wald test on the coefficient of the period of diagnosis, entered as a continuous regressor in a Poisson regression model. (3) Results: The median patient age was stable at 76 years. One-year NS decreased from 83.9% in 1990–2001 to 81.9% in 2009–2015 and 2-year NS from 72.2% to 70.5%. Five|2-year CNS increased from 85.7% to 86.7%. These trends were not significant. In the age stratum 70–79 years, a weakly significant decrease in 2-year NS from 71.4% to 65.7% occurred. Multivariate analysis adjusting for age group at diagnosis and geographic area showed an excess risk of death at 5|2-years, of borderline significance, in 2003–2015 versus 1990–2002. (4) Conclusions: One- and 2-year NS and 5|2-year CNS showed no improvements. Current strategies for VSCC control need to be revised both in Italy and at the global level.

## 1. Introduction

Vulvar squamous cell carcinoma (VSCC) comprises two major aetiologic subtypes. The warty/basaloid type is generally related to the HPV infection, is preceded by usual vulvar intraepithelial neoplasia (VIN) and is primarily detected in younger women. The more common keratinizing type is mostly not related to the HPV infection and occurs in elderly women, often in a background of lichen sclerosus and/or differentiated VIN [1,2].

The incidence of VSCC has increased for decades in a number of Western countries [3,4,5,6,7,8,9]. In a few populations, the rates have followed a stable or non-significantly increasing trend [10]. According to most studies, the incidence increase has been restricted to women aged <50–60 years [3,4,6,7,11,12]. Among older women, in general, the rates have not increased [6,7,12], although with exceptions [4]. These age-related differences are the result of changes in sexual behaviour by recent birth cohorts, with increasing levels of exposure to the human papillomavirus (HPV) infection. In Italy, incidence trends have been partially at variance with this pattern. Between 1990 and 2015, total incidence decreased significantly, which was entirely accounted for by women aged ≥60 years [13]. For younger women, conversely, the incidence rose as expected. Specifically, the risk of VSCC has been increasing for all cohorts born since 1945.

Surgery is the primary treatment for VSCC. Radiation therapy is also given to patients with stage III or IV disease. Newer strategies integrate surgery, radiation therapy, and chemotherapy and tailor the treatment plan to the clinical and pathologic extent of the disease. Patterns of practice in combining these treatment options, however, may vary. There are only limited data on treatment efficacy in advanced disease. As a consequence, there is no standard chemotherapy regimen for these patients [14].

In the Western world, the studies on the prognosis of VSCC have covered time spans of two [4,15,16] to five decades [7] from the 1960s onwards. In general, their results have been unsatisfactory. Between 1989 and 2010, for example, 5-year relative survival in the Netherlands has followed a uniformly stable trend [4]. Five- and 10-year relative survival have also been constant in Japan between 1976 and 2008 [17]. In England, conversely, 1- and 5-year overall survival have increased since 1990, but not significantly so for patients aged 20–39 and 50–59 years [15].

Studies with observation times of greater length have more often documented survival increases that are statistically significant but of modest magnitude. In the Nordic countries, for example, 1- and 5-year relative survival from cumulated vulvar and vaginal cancer have not made consistent improvements for as much as a half century [18]. Only in Norway has 5-year relative survival increased in all age groups, but the starting level was below that of Finland, Sweden, and Denmark. Another 50-year trend study has confirmed, for Norway, a significant increase in 5-year relative survival from VSCC [7].

In other countries, however, the opposite was found. In the U.S. and Canada, for example, a large 4-decade study has provided evidence for a decrease in 2- and 5-year relative survival from VSCC [5]. A recent update has suggested that this unfavourable trend is accounted for by an increased mortality of patients with stage IV tumours [19]. In Australia, the unadjusted 5-year survival from vulvar cancer of all types tended to be lower in 2000–2016 than in 1984–1999. Only after adjustment for tumour stage and other covariates was the decrease no longer significant [20].

The finding of diverging survival trends between high-income countries reflects critical differences in national models of vulvar cancer control and indicates the need that the outcome of patients be assessed at the local population level. In this article, we report on the temporal trends in survival from VSCC in Italy over the last three decades. Previous Italian studies on cancer survival have not taken vulvar cancer into consideration or have pooled patients with vulvar and vaginal cancers [21]. The objective of this article is to bridge this knowledge gap.

## 2. Materials and Methods

### 2.1. Rationale and Design

This study is part of a multistage research project undertaken to explore primarily (1) the trends in incidence of, and in survival from, VSCC in the Italian population [13] and (2) the patterns of diagnosis, stage, and treatment and the outcomes of the disease at two tertiary referral centres in northern Italy [22,23,24,25]. As a related secondary objective, a systematic literature review of epidemiologic studies has recently been carried out [26]. In this article, we report a retrospective, cancer-registry-based, multicentre study of trends in survival of VSCC patients diagnosed in approximately half of the Italian female population between 1990 and 2015.

We explored the time trends in survival using three prognostic indicators, namely: 1- and 2-year net survival (NS) and 5-year conditional NS (CNS). NS is used to estimate the excess mortality due to a given disease when the causes of death of some patients may be missing or, when not, may be inaccurate, unreliable, or highly dependent on the local coding practices [27]. This is especially the case for large cancer registry datasets. NS is defined as the probability to survive cancer in the absence of other causes of death or the survival that would be observed if the disease of interest was the only possible cause of death. Thus, NS is not influenced by cross-sectional differences and temporal changes in mortality from any other cause, which permits unbiased survival comparisons between subpopulations and across time.

It is worth noting that the above three measures of NS inform about the effect of distinct clinical prognostic factors, allowing us to disentangle the early and later survival improvements (if any) over the first five years since diagnosis. One- and 2-year survival are adversely affected by the prevalence of late-stage, rapidly fatal cancers, indicating diagnostic delays or problems with the referral pathway, and their improvements reflect improvements in tumour stage distribution at diagnosis [28]. We defined 5-year CNS as the probability of surviving an additional three years on the condition that the patient has survived two years and in the hypothetical situation in which VSCC was the only possible cause of death. This mid-term outcome measure, hereby referred to as 5|2-year CNS [28], is impacted by more delayed fatalities that are due to the growth of occult micrometastasis at diagnosis and is more sensitive to improvements in adjuvant treatments.

In order to detect subtle changes in survival probability, data analysis was stratified by patient age and geographic area of residence. The former has been commonly reported to be a strong inverse prognostic factor [4,5,7,15,19,20]. The latter influences the prognosis of the greater part of cancers in Italy, with a general pattern of decreasing survival from the north of the country to the south [29].

### 2.2. Source of Data

We used the same dataset as the one we created for a previous study addressing the trends in incidence of the disease in Italy [13]. The data for the study were derived from the database of the Italian Association of Cancer Registries. Thirty-eight provincial/regional population-based registries authorised access to their records. Their geographic distribution is shown in Figure 1.

The study considered the patients diagnosed between 1990 and 2015. As shown in the Appendix A (Appendix A), the participating registries contributed data for periods varying from three to 26 years. It also appears from Appendix A that cancer registries have been introduced in Italy in a phased manner, with a pronounced north–south delay. The median year of registration of VSCC was 2004 for those of northern Italy, 2002 for those of central Italy, and 2007 for those of southern Italy. In 2015, the study covered a total female population of 15,358,161, equivalent to 49.4% of Italian women.

The records of a total of 8347 patients registered with invasive vulvar cancer (topography code C51 according to the International Statistical Classification of Diseases and Related Health Problems, 10th revision) [30] were extracted. Patients aged <15 years, patients with missing follow-up date, cases diagnosed at autopsy, and cases with morphology code other than 8051–8084 (VSCC) (*n* = 2073) were excluded. This left 6274 cases available for analysis. In Italy, information about tumour stage and treatment of vulvar neoplasms is not registered.

### 2.3. Statistical Methods

The study years were divided into three segments, namely: 1990–2001, 2002–2008, and 2009–2015, as obtained by dividing the distribution by year of incidence into tertiles.

Differences in median patient age between time periods were evaluated by means of the Mood’s test, a nonparametric K-sample test for the null hypothesis that the K independent samples were drawn from populations with the same median [31].

One- and 2-year NS rates were calculated using the Pohar-Perme estimator [32]. The estimates, obtained with the *strs* Stata command according to a cohort (or complete) approach [33], were age-standardised using the International Cancer Survival Standard (ICSS)-1 weights [34]. Patients were followed-up until 31 December 2018. To correct for background mortality, administrative region-specific lifetables, published by the Italian National Institute of Statistics, were used.

The 5|2-year CNS with the 95% confidence interval (CI) was obtained from the NS at 2 + 3 years after diagnosis, with the time at risk being computed from two years after diagnosis [35]. The 5|2-year CNS was age-standardised [35] using the ICSS-1 weights [34].

The overall trends in 1- and 2-year NS and 5|2-year CNS were evaluated for total patients and for the subgroups aged <60 (i.e., 15–59) years and <50 (i.e., 15–49) years. Our working hypothesis was that the survival trends might be different in younger patients. To determine the statistical significance of all trends in 1- and 2-year NS and 5|2-year CNS, Poisson regression models were built that included the period of diagnosis as a continuous regressor. Specifically, the statistical significance was assessed with the Wald test for trend, i.e., with the *p*-value and the 95% CI in the exponential of the period of diagnosis coefficient.

The potential bias in survival trends resulting from differences in time periods of cancer registration between southern Italy, where survival from VSCC is lower, and the rest of the country was dealt with through a sensitivity analysis of the age-specific trends in 1- and 2-year NS and 5|2-year CNS. The evaluation was repeated after the exclusion of incidence records (*n* = 1348) obtained from the 15 registries of southern Italy.

Multivariate analysis of 1- and 2-year NS and 5|2-year CNS was performed by calculating the relative excess risk (RER) of death [4]. A flexible parametric survival model using restricted cubic splines was fitted on the log cumulative excess hazard scale by using the *stpm2* Stata command.

All statistical analyses were performed using the Stata statistical package, release 15.1 (StataCorp, College Station, TX, USA).

## 3. Results

### 3.1. Age Distribution

Table 1 shows the distribution of the 6274 study patients by age group at diagnosis. Over two-thirds of them were aged 70 years and above. The median patient age (data not shown) was stable at 76 years in the whole case series, at 75 years among women aged 70–79 years, and at 84 years for those aged ≥80 years. In the age stratum 15–69 years, conversely, there was a significant decrease from 63 years in 1990–2001 to 62 years in 2002–2008 and 61 years in 2009–2015 (*p* < 0.001).

### 3.2. Survival by Age Group and Geographic Area

Table 2 shows 1- and 2-year NS and 5|2-year CNS by patient age and geographic area of residence. Expectedly, both the strong inverse prognostic value of patient age and the survival inequality between the three main geographic areas were confirmed, although the geographic heterogeneity in 1-year NS was of weak statistical significance.

### 3.3. Trends in Survival

Figure 2 depicts the trends in total 1- and 2-year NS and in 5|2-year CNS. Between the first and the last time period, 1- and 2-year NS both decreased by approximately 2%. An increase in 5|2-year CNS of even more modest magnitude was observed. None of these trends was significant. Overall, a little less than 30% patients died in the first two years after diagnosis. Approximately 85% of the remaining survived five years.

The overall trends in Figure 2 were also evaluated among patients aged <60 (i.e., 15–59) years and <50 (i.e., 15–49) years. The absence of significant trends was confirmed (Figure 3).

### 3.4. Trends in Survival by Age Group and Geographic Area

Trends in 1- and 2-year NS and in 5|2-year CNS according to age group and geographic area are shown in Table 3, Table 4 and Table 5, respectively. In none of the strata considered could we detect a significant trend in 1-year NS (Table 3) and in 5|2-year CNS (Table 5). As far as 2-year NS is concerned (Table 4), a statistically significant drop from 71.4% to 65.7% was found in the age stratum 70–79 years. The median patient age in this subgroup was stable at 75 years (*p*-value = 0.248) (data not shown).

For sensitivity analysis purposes, the evaluation of the trends in 1- and 2-year NS and in 5|2-year CNS by age group was repeated after excluding the data obtained from the 15 registries of southern Italy. No significant trend was observed in the rest of the country (data not shown), not even in the 2-year NS of patients aged 70–79 years, who showed a decrease from 71.2% (95% CI, 67.5–74.5%) to 68.9% (95% CI, 63.7–73.5%). In southern Italy, the worsening in 2-year NS in this age group was more pronounced, from 73.0% (95% CI, 59.6–82.6%) in 1990–2001 to 62.9% (95% CI, 55.7–69.2%) in 2002–2008 and 58.7% (95% CI, 50.6–65.9%) in 2009–2015, although the level of significance of this trend was only borderline (*p*-value = 0.101).

### 3.5. Multivariate Analysis of Survival

Table 6, Table 7 and Table 8 show multivariate RER of death at one year and two years since diagnosis and at five years since diagnosis conditional on having survived two years. Patients aged 70–79 years and ≥80 years had a multivariate RER of death at one year significantly higher than the unity, that is, a significant excess mortality from VSCC compared with patients aged 15–69 (treated as a reference category). A moderate excess risk of death was seen in the south of Italy as compared with the north. No change over time was demonstrated. At two years since diagnosis, the pattern of RERs of death was similar. The multivariate RER of death at five years conditional on having survived two years was confirmed both for patients aged 70 years and older and for those living in southern Italy; an excess risk of borderline significance was found in the most recent time period.

## 4. Discussion

### 4.1. Major Findings

This is the first analysis of population-based trends in survival from VSCC ever carried out in Italy. At variance with most other cancer sites [29], we observed no significant changes over the past three decades. Overall, a little less than 30% patients died in the first two years after diagnosis, and approximately 85% of the remaining patients survived five years. A worsening of 2-year NS was observed among patients aged 70–79 years. This decreasing trend was mainly accounted for by the data registered in southern Italy.

Multivariate analysis confirmed the stagnation of survival outcomes. Comparing the years after 2000 versus the 1990s, we found an excess risk of death at 5|2-years that was adjusted for patient age as well as geographic area. This result had a borderline level of significance.

### 4.2. Interpretation of Results

Though somewhat expected based on the international literature [4,5,7,15,16,17,18,19,20], these results are nonetheless disappointing because population-based cancer survival is a measure of the overall effectiveness of the research system and the healthcare system in dealing with the clinical management of VSCC [36]. It appears that there are some chronic problems with this disease for which a solution seems still to be remote [37]. First, there remains a substantial clinical research gap to be filled. Pharmaceutical companies have little or no interest in developing and marketing drugs for rare diseases. Conducting clinical trials for these conditions poses many challenges such as, for example, the need to open multiple recruitment sites (including inexperienced sites) coupled with differences in national clinical trial regulations [38]. In fact, since scientific works on rare diseases are poorly financed, these efforts are difficult to sustain.

Second, tumour stage distribution of vulvar cancer at diagnosis has not substantially changed for decades in Italy as in other countries [4,7,39]. In one of our previous studies, which was based in northern Italy (like most of cancer registries providing data for the present analysis), we explored the time trend in the probability of lymph node involvement and in the number of positive nodes [25]. Over a 4-decade period, both showed no significant changes. From this perspective, the finding that 2-year NS of patients aged 70–79 years has been decreasing is particularly concerning, as it can be interpreted to suggest an increase in the proportion of patients presenting with late-stage diseases that are fatal in a relatively short term [40]. The persistent inability to detect the disease at a more curable stage reflects the unavailability of appropriate screening techniques but also indicates the ineffectiveness of current referral systems, where communication and exchange of experience between tertiary-level centres and primary/secondary care levels are insufficient.

A third factor to consider is the aetiology of the disease. Since patients with HPV-dependent VSCC are at lower risk of recurrence and death [41], the hypothesis has been raised [7] that the slowly increasing trend in survival reported from some Nordic countries [18] may result not from advances in diagnosis and treatment but instead from an increasing incidence of HPV-positive vulvar cancer—a trend that is well documented [42]. Following this line of reasoning, Italian patients with VSCC—at least the aged ones—are probably disadvantaged by still having a low prevalence of HPV. Approximately 90% of VSCC cases in Italy are diagnosed among women ≥60 years. As we have previously shown, the incidence of the disease in this subset of the female population has been decreasing between 1990 and 2015 [13]. The reason is that the earliest cohorts in our study had experienced a downward trend in exposure to the HPV infection as a result of the dramatic socio-economic improvement that occurred soon after World War II. Only for subsequent generations, an opposite trend is currently observed. This is the most likely explanation for the decrease in median patient age in the age stratum 15–69 years, which was not observed at 70 years and above. It clearly appears, however, that the aetiologic evolution is still too weak and limited to influence the overall outcome of the disease.

The stability of survival from vulvar cancer may be considered a favourable result only from the perspective that there is currently a trend toward less extensive vulvar surgery [43] and increasing use of the sentinel lymph node biopsy in tumours <4 cm in size [44] and of the radio–chemo therapeutic approach in late-stage disease [45]. This was the conclusion, for example, of a U.S. study reporting a stable trend in 5-year overall and disease-specific survival from stage III–IV vulvar cancer between 1988 and 2007 [39]. An evolution towards less aggressive surgery has also been reported from Italy [46].

### 4.3. Policy Implications

If 1- and 2-year NS and 5|2-year CNS data are considered together, this study clearly suggests that most VSCC deaths in Italy are due to an advanced stage of disease at the time of diagnosis, since approximately 30% patients die in the subsequent two years and approximately 85% of the remaining survive five years. By implication, the primary need is to improve vulvar cancer care at the community level, in particular to detect the disease earlier than is currently the case. To this end, we have previously proposed the adoption of a hub-and-spoke organisation [25]. This model would facilitate the exchange of experience, a continuing communication, the creation of a common knowledge base, the standardisation of referral guidelines, and the communication and cooperation between gynaecologists, dermatologists, and pathologists. Cooperation between gynaecologists and dermatologists at the community level, too, would be a valuable resource.

There are complementary actions that can be taken. It is of the utmost importance, in particular, to perform a correct vulvar inspection during the diagnostic work-up of women with abnormal Pap smear results.

Vulvar cancer screening by inspection has been proposed by some [47]. Medical societies, however, do not routinely recommend an external genital examination in women aged 65 years and above. There are no well-defined screening protocols and both large-scale technical feasibility and economic viability of this approach—given the low prevalence of disease—are doubtful. With respect to self-examination, the benefit still awaits evaluation [48].

Women treated for cervical intraepithelial neoplasia have a considerably higher risk of being later diagnosed with cervical and other HPV-related cancers, including vulvar cancer [49]. The excess risk is higher for women aged >50 years at baseline. Prolonged surveillance of these patients is encouraged, in that it might lead to the early detection of multiple malignancies.

According to European and international guidelines, non-HPV-related VSCC could be prevented by accurate diagnosis, treatment, and follow up of vulvar lichen sclerosus, lichen planus, and differentiated VIN [50,51]. Women with lichen sclerosus are at increased risk of VSCC especially if aged ≥70 years at baseline [52].

Despite the absence of controlled clinical data, prophylactic HPV vaccination will most likely have an impact on VSCC rates [53]. Recently, data from Denmark and the U.S. have shown an ecological association between the introduction of HPV vaccination and a downturn in incidence rates of vulvar cancer and precancerous lesions among women aged <20 years, 20–29 years [54], and 20–44 years [55]. Among older patients, HPV infection accounts for a minority of VSCC cases [56], which is smaller in Italy than in other Western countries [13]. By implication, the effect of vaccination on incidence is expected to be more limited.

With respect to treatment, the challenges facing the treatment of vulvar malignancies can be addressed by changes in regulations for drug development and by national and international initiatives to promote the clinical research in the field of rare diseases [38,57,58]. There are promising lines of research, such as the investigation of potentially targetable markers (for example, the prostaglandin E2 receptor 4) [59] and of new potential immunotherapeutic approaches to the management of the disease (for example, the use of state-of-the-art immune checkpoint inhibitors) [60].

### 4.4. Strengths and Weaknesses

The large size of this study, covering half of Italian women, has limited the adverse statistical implications of the low absolute incidence of VSCC, which requires a very large sample to reject the null hypothesis of no changes in survival.

This strength, however, is counterbalanced by closely related limitations. First, cancer registration in Italy has been introduced at the provincial/regional level since the 1970s but with irregularities in time and space, which have been suggested to cause biases in the results of trend studies [13,61]. In particular, the registries of southern Italy, where survival from VSCC is lower, have generally been established later than elsewhere. For this reason, we performed a sensitivity analysis of the age-specific trends in 1- and 2-year NS and in 5|2-year CNS by excluding the data from the 15 registries of southern Italy.

Second, cancer registries in Italy and elsewhere do not routinely collect sufficient patient information, particularly on disease risk factors, socioeconomic and demographic characteristics, and clinical characteristics, including tumour stage, treatment, comorbidities, and disease recurrences. The use of 1- and 2-year NS, however, allowed us to evaluate the prognostic impact of advanced vulvar cancers despite the lack of specific information on tumour stage at diagnosis.

Even though 2-year NS has already been used in other studies on VSCC [5,19], its meaning needs an explanation. We consider this outcome measure to convey similar information to 1-year NS because 2-year NS, too, is influenced by the prevalence of late-stage cancers. As VSCC tends to metastasize not through the blood vessels but mainly through the lymphatic vasculature and through local invasion, advanced diseases may not be as rapidly fatal as, for example, a stage IV lung cancer. Even patients with FIGO stage IVB VSCC die during the second years after diagnosis in a considerable proportion of cases [62].

## 5. Conclusions

One- and 2-year NS and 5|2-year CNS of patients with VSCC in Italy have not improved since 1990, which confirms the frustrating data previously reported from several other countries. Current strategies for the control of VSCC need to be reconsidered and substantially revised both in Italy and at the global level. The main proposed actions include promoting the clinical research on the disease, adopting a hub-and-spoke model of care, performing vulvar inspection in cervical cancer screening activities and undertaking regular follow-ups of women at high risk. HPV vaccination is expected to prevent a considerable proportion of vulvar cancers in younger women.

## Figures and Tables

**Figure 1 jcm-12-02172-f001:**
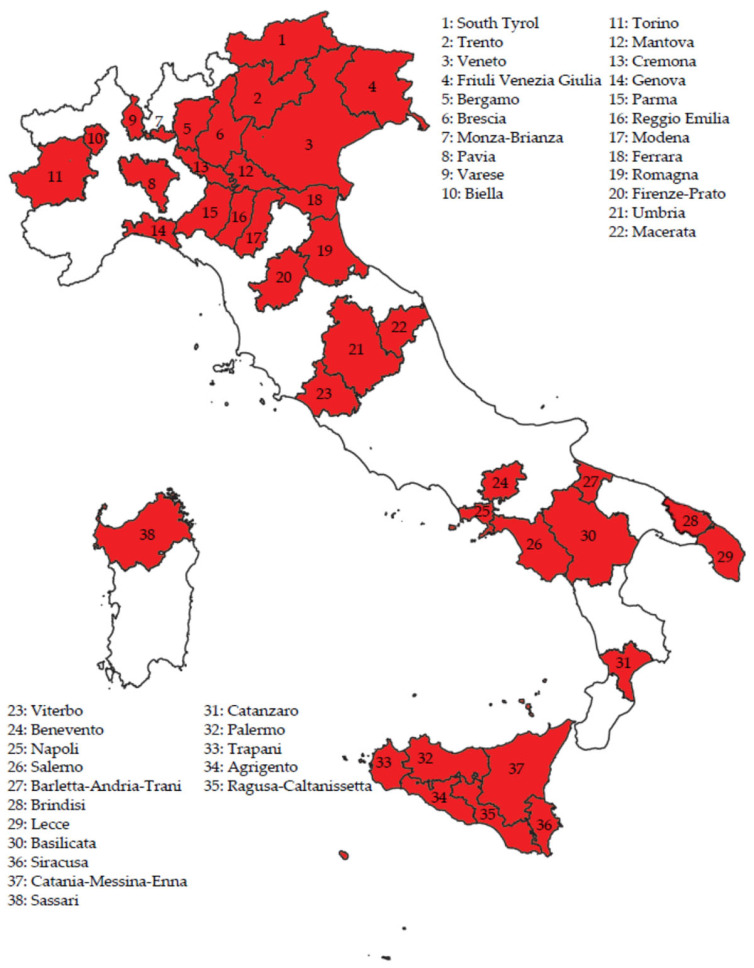
Geographic distribution of the 38 cancer registries participating in the study on time trends in net survival from vulvar squamous cell carcinoma in Italy between 1990 and 2015 (total resident female population 15,358,161 in 2015).

**Figure 2 jcm-12-02172-f002:**
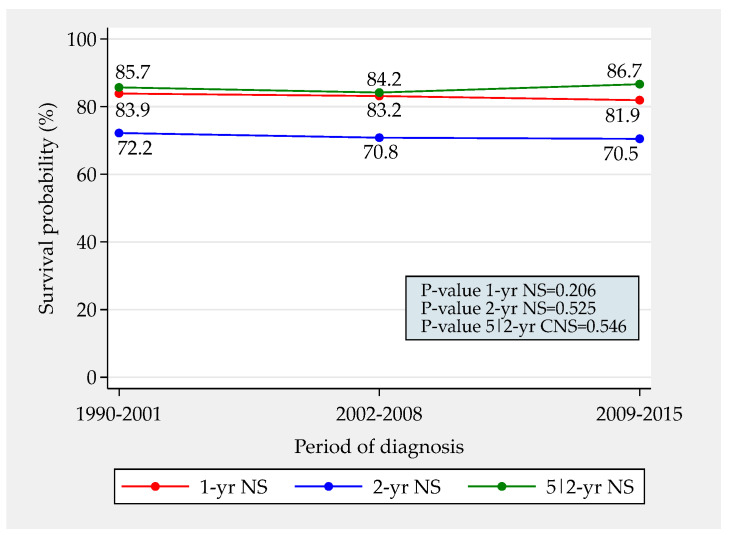
Trends in total 1- and 2-year net survival and in 5|2-year conditional net survival from vulvar squamous cell carcinoma in Italy between 1990 and 2015 (total resident female population 15,358,161 in 2015).

**Figure 3 jcm-12-02172-f003:**
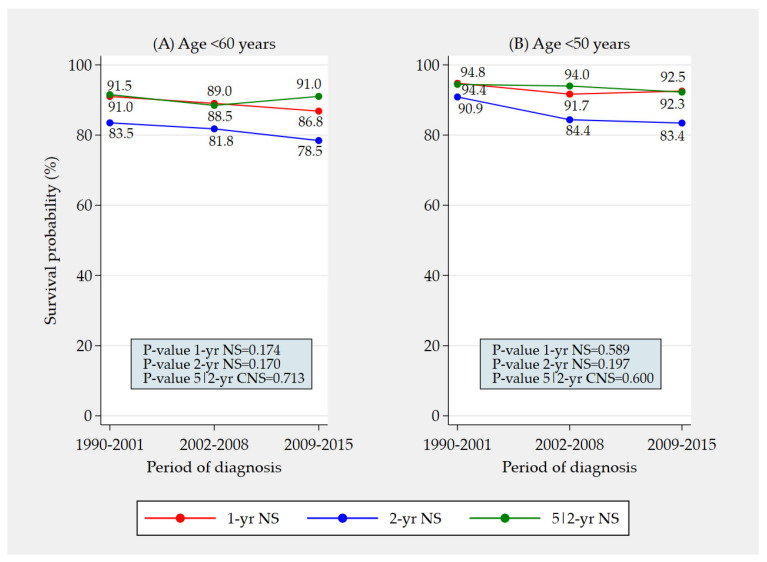
Trends in total 1- and 2-year net survival and in 5|2-year conditional net survival from vulvar squamous cell carcinoma in Italy between 1990 and 2015 among patients aged <60 (i.e., 15–59) years (**A**) and <50 (i.e., 15–49) years (**B**) (total resident female population 10,715,318 aged <60 years and 8,537,374 aged <50 years in 2015).

**Table 1 jcm-12-02172-t001:** Number of patients registered with vulvar squamous cell carcinoma by age group at diagnosis, geographic area, and time period. Italy, 1990–2015 (total resident female population 15,358,161 in 2015).

	Time Period			Total
	1990–2001	2002–2008	2009–2015	
Age group				
15–69 years	620 (27.3)	647 (28.9)	505 (28.6)	1772 (28.2)
70–79 years	834 (36.7)	810 (36.2)	568 (32.2)	2212 (35.3)
≥80 years	817 (36.0)	783 (35.0)	690 (39.1)	2290 (36.5)
Geographic area				
North	1653 (72.8)	1365 (60.9)	1056 (59.9)	4074 (64.9)
Centre	407 (17.9)	265 (11.8)	180 (10.2)	852 (13.6)
South	211 (9.3)	610 (27.2)	527 (29.9)	1348 (21.5)
Total	2271 (100.0)	2240 (100.0)	1763 (100.0)	6274 (100.0)

Numbers in parentheses are column percentages. Total percentages may not sum to 100.0% due to rounding.

**Table 2 jcm-12-02172-t002:** One- and 2-year net survival and 5|2-year conditional net survival from vulvar squamous cell carcinoma by age group at diagnosis and geographic area. Italy, 1990–2015 (total resident female population 15,358,161 in 2015).

	1-Year Net Survival, % (95% CI) ^§^	2-Year Net Survival, % (95% CI) ^§^	5|2-Year Conditional Net Survival	
			No. at Risk	% (95% CI)
Total	83.0 (82.0–84.1)	71.2 (69.9–72.5)	3733	85.4 (83.8–86.9)
Age group *				
15–69 years	86.5 (84.9–88.1)	75.6 (73.5–77.6)	1299	87.7 (85.6–89.6)
70–79 years	81.4 (79.7–83.1)	68.8 (66.8–70.9)	1428	83.3 (80.6–85.7)
≥80 years	71.8 (69.8–74.0)	53.8 (51.4–56.3)	1006	78.4 (72.9–82.9)
*p*-value ^†^	<0.001	<0.001		<0.001
Geographic area ^‡^				
North	83.7 (82.4–85.0)	72.6 (71.0–74.2)	2459	87.5 (85.6–89.2)
Centre	82.9 (79.9–86.1)	70.3 (66.4–74.3)	513	82.6 (77.6–86.6)
South	81.0 (78.7–83.4)	67.2 (64.4–70.1)	761	80.4 (76.4–83.8)
*p*-value ^†^	0.083	0.001		<0.001

CI, confidence interval. * Age-specific 1- and 2-year net survival and 5|2-year conditional net survival are shown. ^†^
*p*-values, for the Wald test for survival comparisons, refer to the variable’s coefficient estimated by fitting a generalised linear model for survival estimates with a Poisson distribution, including the follow-up time, the age at diagnosis and the covariate. The cohort approach was used. ^‡^ One- and 2-year net survival and 5|2-year conditional net survival are shown, all age-standardised using the International Cancer Survival Standard-1 weights. ^§^ Numbers at risk are provided in the ‘Total’ column in Table 1.

**Table 3 jcm-12-02172-t003:** Trend in 1-year net survival from vulvar squamous cell carcinoma by age group at diagnosis and geographic area. Italy, 1990–2015 (total resident female population 15,358,161 in 2015).

	1-Year Net Survival, % (95% CI) ^‡^			*p*-Value ^§^
	1990–2001	2002–2008	2009–2015	
Age group *				
15–69 years	86.4 (83.4–88.9)	87.0 (84.1–89.4)	85.9 (82.5–88.7)	0.810
70–79 years	82.1 (79.1–84.7)	82.2 (79.2–84.8)	79.2 (75.5–82.4)	0.235
≥80 years	73.6 (69.9–76.9)	69.4 (65.6–72.9)	72.5 (68.5–76.1)	0.486
Geographic area ^†^				
North	84.2 (82.1–86.1)	84.0 (81.7–86.1)	82.6 (79.8–85.0)	0.427
Centre	83.1 (78.2–87.0)	83.6 (76.8–88.5)	82.9 (75.7–88.2)	0.310
South	81.9 (75.6–86.6)	81.2 (77.5–84.3)	80.8 (76.7–84.2)	0.700

CI, confidence interval. * Time trends in age-specific 1-year net survival are shown. ^†^ Time trends in age-standardised (International Cancer Survival Standard-1 weights) 1-year net survival are shown. ^‡^ Numbers at risk are provided in the ‘Total’ column in Table 1. ^§^
*p*-values are for the Wald test for trend in the exponential of the period of diagnosis coefficient entered as a continuous regressor in a Poisson regression model for net survival.

**Table 4 jcm-12-02172-t004:** Trend in 2-year net survival from vulvar squamous cell carcinoma by age group at diagnosis and geographic area. Italy, 1990–2015 (total resident female population 15,358,161 in 2015).

	2-Year Net Survival, % (95% CI) ^‡^			*p*-Value ^§^
	1990–2001	2002–2008	2009–2015	
Age group *				
15–69 years	74.9 (71.2–78.2)	75.3 (71.7–78.5)	76.7 (72.6–80.2)	0.628
70–79 years	71.4 (67.9–74.6)	68.4 (64.8–71.7)	65.7 (61.4–69.7)	0.037
≥80 years	52.8 (48.6–56.9)	53.1 (48.8–57.2)	55.6 (51.0–59.9)	0.606
Geographic area ^†^				
North	73.0 (70.4–75.4)	72.4 (69.5–75.1)	72.3 (69.1–75.3)	0.842
Centre	71.2 (65.3–76.2)	68.0 (60.0–74.8)	69.3 (59.5–77.2)	0.519
South	69.6 (62.5–75.5)	67.2 (62.9–71.1)	67.0 (62.2–71.4)	0.708

CI, confidence interval. * Time trends in age-specific 2-year net survival are shown. ^†^ Time trends in age standardised (International Cancer Survival Standard-1 weights) 2-year net survival are shown. ^‡^ Numbers at risk are provided in the ‘Total’ column in Table 1. ^§^
*p*-values are for the Wald test for trend in the exponential of the period of diagnosis coefficient entered as a continuous regressor in a Poisson regression model for net survival.

**Table 5 jcm-12-02172-t005:** Trend in 5|2-year conditional net survival from vulvar squamous cell carcinoma by age group at diagnosis and geographic area. Italy, 1990–2015 (total resident female population 15,358,161 in 2015).

	5|2-Year Conditional Net Survival						*p*-Value ^§^
	1990–2001		2002–2008		2009–2015		
	No. at Risk	% (95% CI)	No. at Risk	% (95% CI)	No. at Risk	% (95% CI)	
Age group *							
15–69 years	457	87.9 (84.2–90.8)	482	85.4 (81.7–88.5)	360	91.8 (87.7–94.6)	0.351
70–79 years	562	82.2 (77.7–85.8)	525	84.1 (79.7–87.7)	341	84.2 (77.9–88.8)	0.694
≥80 years	349	81.8 (71.1–88.8)	343	75.6 (66.3–82.7)	314	76.5 (65.6–84.3)	0.654
Geographic area ^†^							
North	994	87.7 (84.5–90.3)	834	86.6 (83.2–89.3)	631	88.5 (84.3–91.7)	0.434
Centre	250	81.1 (73.2–86.8)	169	84.0 (73.3–90.7)	94	82.1 (68.6–90.1)	0.736
South	124	80.5 (69.1–88.1)	347	79.3 (73.5–84.0)	290	83.0 (76.0–88.2)	0.224

CI, confidence interval. * Time trends in age-specific 5|2-year conditional net survival are shown. ^†^ Time trends in age-standardised (International Cancer Survival Standard-1 weights) 5|2-year conditional net survival are shown. ^§^
*p*-values are for the Wald test for trend in the exponential of the period of diagnosis coefficient entered as a continuous regressor in a Poisson regression model for conditional net survival.

**Table 6 jcm-12-02172-t006:** Multivariate relative excess risk (RER) of death from vulvar squamous cell carcinoma at one year since diagnosis by age group at diagnosis, geographic area, and period of diagnosis. Italy, 1990–2015 (total resident female population 15,358,161 in 2015).

	Deaths, *n* (%)	RER (95% CI) *	*p*-Value ^†^
Age group			<0.001
15–69 years	248 (14.0)	1.00 (ref)	
70–79 years	453 (20.5)	1.43 (1.21–1.69)	
≥80 years	790 (34.5)	2.28 (1.94–2.67)	
Geographic area			0.096
North	954 (23.4)	1.00 (ref)	
Centre	204 (23.9)	1.04 (0.87–1.24)	
South	333 (24.7)	1.18 (1.02–1.37)	
Period of diagnosis			0.341
1990–2002	581 (23.1)	1.00 (ref)	
2003–2015	910 (24.2)	1.06 (0.94–1.21)	

CI, confidence interval. * Estimates were performed adjusting for all variables in the Table. The RER of death is from a flexible parametric model for net survival with three knots, i.e., the model with the lowest Akaike information criterion. ^†^
*p*-values are for the Wald test.

**Table 7 jcm-12-02172-t007:** Multivariate relative excess risk (RER) of death from vulvar squamous cell carcinoma at two years since diagnosis by age group at diagnosis, geographic area, and period of diagnosis. Italy, 1990–2015 (total resident female population 15,358,161 in 2015).

	Deaths, *n* (%)	RER (95% CI) *	*p*-Value ^†^
Age group			<0.001
15–69 years	447 (25.2)	1.00 (ref)	
70–79 years	759 (34.3)	1.36 (1.19–1.54)	
≥80 years	1267 (55.3)	2.23 (1.97–2.52)	
Geographic area			0.001
North	1581 (38.8)	1.00 (ref)	
Centre	330 (38.7)	0.99 (0.86–1.14)	
South	562 (41.7)	1.24 (1.11–1.40)	
Period of diagnosis			0.932
1990–2002	990 (39.4)	1.00 (ref)	
2003–2015	1483 (39.4)	1.00 (0.90–1.10)	

CI, confidence interval. * Estimates were performed adjusting for all variables in the Table. The RER of death is from a flexible parametric model for net survival with eight knots, i.e., the model with the lowest Akaike information criterion. ^†^
*p*-values are for the Wald test.

**Table 8 jcm-12-02172-t008:** Multivariate relative excess risk (RER) of death from vulvar squamous cell carcinoma at five years since diagnosis conditional on having survived two years, by age group at diagnosis, geographic area, and period of diagnosis. Italy, 1990–2015 (total resident female population 15,358,161 in 2015).

	Deaths, *n* (%)	RER (95% CI) *	*p*-Value ^†^
Age group			<0.001
15–69 years	462 (35.6)	1.00 (ref)	
70–79 years	842 (59.0)	1.33 (1.11–1.60)	
≥80 years	747 (74.3)	1.55 (1.20–1.99)	
Geographic area			0.003
North	1362 (55.4)	1.00 (ref)	
Centre	325 (63.4)	1.10 (0.86–1.40)	
South	364 (47.8)	1.43 (1.17–1.74)	
Period of diagnosis			0.073
1990–2002	1147 (75.7)	1.00 (ref)	
2003–2015	904 (40.8)	1.18 (0.98–1.42)	

CI, confidence interval. * Estimates were performed adjusting for all variables in the Table. The RER of death is from a flexible parametric model for net survival with six knots, i.e., the model with the lowest Akaike information criterion. ^†^
*p*-values are for the Wald test.

## Data Availability

The anonymised dataset used in this study is available on request from the corresponding author.

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
