# Peer review of "Trends in Net Survival from Vulvar Squamous Cell Carcinoma in Italy (1990–2015)"

_jcm, 2023, doi:10.3390/jcm12062172_

Round 1

Reviewer 1 Report

Dear Authors 

This is a very interesting and well-written article. However, you have to pay attention to some points:

Αdd the purpose of your study to the abstract section. 

Υour research background is very little. You should perhaps reinforce this with the definition of the disease, its stages, epidemiology, and then treatment.

The purpose of your study should always be included at the end of the introduction. 

In the methodology section, specify exactly the type of study (e.g. retrospective). 

Restate any important facts in this post. What was the study population?

Also, each table must have the number of the population

Were there data on factors influencing mortality? What were the deaths related to?

Among the limitations of the study is its design. It is important to conduct large population-based cohorts investigating mortality factors, demographics beyond age, and stage of cancer at diagnosis. 

Author Response

Reviewer 1

Αdd the purpose of your study to the abstract section.

Response. I have added the statement that the study was aimed at bridging the knowledge gap about vulvar cancer survival in Italy (lines 74-76).

Υour research background is very little. You should perhaps reinforce this with the definition of the disease, its stages, epidemiology, and then treatment.

Response. I have added this background information (lines 92-113).  

The purpose of your study should always be included at the end of the introduction.

Response. I have re-written the last paragraph of the Introduction. I have stated that the objective of the study was to bridge the knowledge gap about survival from vulvar cancer in Italy (lines 138-141).

In the methodology section, specify exactly the type of study (e.g. retrospective). Restate any important facts in this post. What was the study population?

Response. In the first paragraph of the Methods section, I have added a detailed definition of the study design (lines 148-151).

Also, each table must have the number of the population

Response. I have provided the number of the population in all Table titles and Figure legends: (total resident female population 15,358,161 in 2015). In Figure 3, I have indicated the number of women aged <60 years and <50 years because the graphs consider only these subgroups of the female population. 

Were there data on factors influencing mortality? What were the deaths related to?

Response. I have added a multivariate flexible parametric survival model in order to evaluate the available prognostic factors (lines 230-233; lines 320-330; Tables 6 to 8). With respect to deaths, it must be noted that net survival, usually used in cancer-registry-based studies, may be unfamiliar to a clinically-oriented reader. Thus, I have expanded the definition of net survival in the Methods section (lines 153-157). In brief, net survival allows to estimate the excess mortality due to a given disease when the causes of death are not reliable. This is the case for large cancer registry data sets. In brief, all deaths in the study should be assumed to be due to vulvar cancer, although their occurrence is based on an estimate.

Among the limitations of the study is its design. It is important to conduct large population-based cohorts investigating mortality factors, demographics beyond age, and stage of cancer at diagnosis.

Response. I agree that a research effort with these objectives would be needed. A critical problem, however, is that cancer registries do not routinely collect sufficient patient information on risk factors, socioeconomic and demographic characteristics and clinical characteristics (tumour stage, treatment, comorbidities and disease recurrences). This key limitation of the study is now acknowledged (lines 467-472). Please note that the article explains (lines 470-472) that the use of 1- and 2-year survival allows to evaluate the prognostic impact of advanced cancers despite the lack of information on tumour stage. 

Reviewer 2 Report

The authors prepared an interesting population based study exploring the survival of patients with vulvar cancer. The study included almost half of Italian female population with more than 6000 vulvar cancer patients. Congratulations to the authors for huge amount of work. The results are in line with already published and in line with the methodology used. Authors also sufficiently describe the weaknesses of the study.

I would only suggest to consider rewriting of discussion and conclusions in regard to the suggested actions by the authors. First of all I am missing the discussion and conclusions in regard to HPV vaccination, as this is probably the most efficient action to prevent vulvar cancers, associated with HPV. A meassure that may prevent illness. Second, there are no high quality evidence that screening is beneficial. If authors know strong evidence for it, it should be presented and discussed. In conclusion authors suggest regular follow up of women at high risk. Authors should include the details of what is defined as ''high risk'' and how to do the regular follow up. We should be careful not to suggest something that may not be beneficial or may be associated with high burden for the health system with no impact on survival. Maybe authors could include recommendations of different international societies on prevention, diagnosis and treatment of vulvar cancer patients and compare those guidelines to the reality in Italy.

Author Response

Reviewer 2

I would only suggest to consider rewriting of discussion and conclusions in regard to the suggested actions by the authors. First of all, I am missing the discussion and conclusions in regard to HPV vaccination, as this is probably the most efficient action to prevent vulvar cancers, associated with HPV. A measure that may prevent illness.

Response. This is correct. I have added a paragraph in which recent studies reporting an ecological association between the introduction of HPV vaccination and a decrease in incidence rates of vulvar cancer among younger women is mentioned (lines 441-448). 

Second, there are no high quality evidence that screening is beneficial. If authors know strong evidence for it, it should be presented and discussed.

Response. I agree that there is no evidence for the effectiveness of screening. I have rewritten the paragraph (lines 426-431). Medical societies do not recommend an external genital examination in women aged 65 years and above. Also, there are no well-defined screening protocols, and technical feasibility of this approach is doubtful.   

In conclusion authors suggest regular follow up of women at high risk. Authors should include the details of what is defined as ''high risk'' and how to do the regular follow up. We should be careful not to suggest something that may not be beneficial or may be associated with high burden for the health system with no impact on survival.

Response. I have more precisely identified the patients with high-risk conditions for whom clinical experts suggest more clinical attention, namely: women treated for cervical intraepithelial neoplasia and patients with lichen sclerosus, lichen planus and vulvar intraepithelial neoplasia (lines 432-440). I have provided the necessary references.   

Maybe authors could include recommendations of different international societies on prevention, diagnosis and treatment of vulvar cancer patients and compare those guidelines to the reality in Italy.

Response. This would be an important but challenging topic to evaluate in future research. At present, there are only anecdotal data on practice patterns for vulvar cancer care in Italy. In this manuscript, I can only indicate the international recommendations for the surveillance of high-risk patient (lines 432-440) and disease treatment (lines 108-113).

Round 2

Reviewer 1 Report

Dear Authors

You made a really hard effort to revise this manuscript. Congratulations!

I wish you good luck!